# Characteristics of Patients with Heart Failure and Advanced Chronic Kidney Disease (Stages 4–5) Not Undergoing Renal Replacement Therapy (ERCA-IC Study)

**DOI:** 10.3390/jcm12062339

**Published:** 2023-03-17

**Authors:** Sandra Valdivielso Moré, Miren Vicente Elcano, Anna García Alonso, Sergi Pascual Sanchez, Isabel Galceran Herrera, Francesc Barbosa Puig, Laia C. Belarte-Tornero, Sonia Ruiz-Bustillo, Ronald O. Morales Murillo, Clara Barrios, Joan Vime-Jubany, Nuria Farre

**Affiliations:** 1Heart Failure Unit, Department of Cardiology, Hospital del Mar, 08003 Barcelona, Spain; 2Biomedical Research Group on Heart Disease, Hospital del Mar Medical Research Group (IMIM), 08003 Barcelona, Spain; 3Department of Medicine, Universitat Autónoma de Barcelona, 08193 Barcelona, Spain; 4Department of Nephrology, Hospital del Mar, 08003 Barcelona, Spain; 5Department of Medicine, Universidad Pompeu Fabra, 08002 Barcelona, Spain

**Keywords:** advanced chronic kidney disease, cardiorenal, heart failure, kidney dysfunction, prognosis, mortality

## Abstract

Background: Despite the frequent coexistence of heart failure (HF) in patients with advanced chronic kidney disease (CKD), it has been understudied, and little is known about its prevalence and prognostic relevance. Methods: **A** retrospective study of 217 patients with advanced CKD (stages 4 and 5) who did not undergo renal replacement therapy (RRT). The patients were followed up for two years. The primary outcome was all-cause death or the need for RRT. Results: Forty percent of patients had a history of HF. The mean age was 78.2 ± 8.8 years and the mean eGFR was 18.4 ± 5.5 mL/min/1.73 m^2^. The presence of previous HF identified a subgroup of high-risk patients with a high prevalence of cardiovascular comorbidities and was significantly associated with the composite endpoint of all-cause hospitalization or need for RRT (66.7% vs. 53.1%, HR 95% CI 1.62 (1.04–2.52), *p* = 0.034). No differences were found in the need for RRT (27.6% vs. 32.2%, *p* = 0.46). Nineteen patients without HF at baseline developed HF during the follow-up and all-cause death was numerically higher (36.8 vs. 19.8%, *p* = 0.1). Conclusions: Patients with advanced CKD have a high prevalence of HF. The presence of previous HF identified a high-risk population with a worse prognosis that required close follow-up.

## 1. Introduction

Chronic kidney disease (CKD) and heart failure (HF) are prevalent diseases with high morbidity and mortality rates [1]. Both entities share a common burden of traditional cardiovascular risk factors, such as hypertension and diabetes, which are known to cause or worsen CKD and HF [2]. Moreover, the presence of one of the two may precipitate or exacerbate the other [3,4]. This has led to the concept of the cardio-renal syndrome [5]. The cardio-renal syndrome is a term used to describe a condition in which there is an intricate interplay between the heart and the kidneys, resulting in one or both organs being affected. It is classified into five types of cardio-renal syndrome, depending on which organ causes the initial damage: Type 1 and Type 2 in heart disease; Type 3 and 4 in kidney disease; and Type 5, where both coexist, especially in patients with diabetes mellitus. However, this distinction between the different types of cardio-renal syndrome can be challenging and, frequently, clinically irrelevant [6,7]. The high burden of shared cardiovascular disease and the clear worsening of the patient’s prognosis when both organs are affected make joining efforts in a multidisciplinary approach for cardiorenal patients mandatory.

Extensive evidence shows that CKD is frequent in HF and associated with a worse prognosis in both acute and chronic HF [8,9]. However, most studies have focused on patients with CKD in stages 1–3. Hence, information about advanced CKD (stages 4–5) in HF is scarce. Conversely, studies on CKD looking at HF also focus on the less severe CKD stages (mean estimated glomerular filtration rate (eGFR) < 60 mL/min/1.73 m^2^) [3] or in advanced CKD already in renal replacement therapies [10]. Therefore, little is known about the interaction between HF and advanced CKD in patients not receiving renal replacement therapies. Although the percentage of patients with both HF and advanced CKD is low, the absolute number of patients is not irrelevant. Moreover, this group of patients poses a clinical challenge because most of the medications studied and approved for the treatment of HF are limited to patients with CKD in stages 1 to 3b (i.e., eGFR ≥ 30 mL/min/1.73 m^2^) [11]. 

Therefore, this study aimed to analyze the prevalence of HF in patients with advanced renal disease (stages 4 and 5) and to assess whether the presence of HF conferred a worse prognosis than in patients without HF.

## 2. Materials and Methods

This was a retrospective analysis of adult patients with advanced CKD (stages 4 and 5) who were followed up at the Nephrology outpatient clinic. We included all patients with an estimated glomerular filtration rate (eGFR) below 30 mL/min/1.73 m^2^ CKD-EPI formula. 

Patients were included from the Nephology outpatient clinic on 1 January 2020 and were followed up until 31 December 2021, or they died. We excluded patients on RRT programs or who had undergone a kidney transplant. Follow-up and treatment were performed according to local protocols and the treating physician’s criteria. The chronic kidney was classified as stage 4 Severe CKD (GFR = 15–29 mL/min/1.73 m^2^) and stage 5 End Stage CKD (GFR < 15 mL/min/1.73 m^2^) [12]. Heart failure diagnosis was made according to the European Society of Cardiology HF guidelines [11]. HF with preserved ejection fraction (EF) (HfpEF) was defined as an EF greater than or equal to 50%, HF with reduced EF (HfrEF) as an EF less than or equal to 40%, and HF with mildly reduced EF (HFmrEF) as an EF between 41% and 49%.

At the inclusion date, we analyzed baseline characteristics, including cardiovascular risk factors, comorbidities, cardiac and renal history, baseline laboratory tests, and medical treatment. If available, we collected the presence of previous heart failure (HF) and ejection fraction data from the most recent echocardiogram.

The primary endpoint was to analyze the clinical differences and outcomes between patients with advanced CKD with and without a history of HF. The primary outcome was all-cause mortality or the need for renal replacement therapy. Our secondary endpoints were to characterize the patients with HF history and to analyze the baseline characteristics and outcomes of patients without HF at baseline but who developed HF during the follow-up.

Due to its retrospective nature, the local Ethics Committee (CEIm number 2021/10008) approved the study and waived the need for written informed consent.

### Statistical Analysis

Continuous variables are described as the mean and standard deviation and, when they do not follow a normal distribution, as the median and the 25–75 percentile, and categorical variables as frequency and percentage. Clinical differences were analyzed using the X^2^ test and Fisher’s exact test when appropriate for qualitative variables. Continuous variables were analyzed using the unpaired *t*-student test or Mann–Whitney U test, as appropriate. A Cox proportional hazard model was developed, adjusting for age, sex, diabetes, previous myocardial infarction, atrial fibrillation, valve disease, history of HF, eGFR, use of loop diuretic, beta-blockers and RAS inhibitor (ACEi, ARB2, INRA). A simultaneous adjustment was chosen for all variables included in the model through the Enter procedure. These variables were selected either because they presented statistically significant differences in the bivariate analysis or because they had been identified as potential confounding factors according to the literature. NT-proBNP levels were not included because of the higher number of missing data. Kaplan–Meier curves were constructed to compare the results between patients who presented decompensated HF and those who did not. The results were expressed in the hazard ratio (HR) with a confidence interval of 95%. Statistical significance was set at *p* < 0.05. Statistical analysis was performed with SPSS version 22.0 software (SPSS, IBM, Chicago, IL, USA).

## 3. Results

### 3.1. Baseline Characteristics

A total of 217 patients with advanced renal disease (stages 4 and 5) were included. Eighty-seven patients (40%) had a previous diagnosis of HF (Table 1). These patients were older, more frequently female, and diabetic. There was a high prevalence of cardiovascular risk factors in both groups. The mean eGFR was 18.0 ± 5.5 mL/min/1.73 m^2^, and 73.3% of patients were on stage 4, without difference between groups. Non-cardiac comorbidities were not different between groups, but cardiac comorbidities were significantly more frequent in patients with a history of HF.

The median time from diagnosis of heart failure to inclusion in the study was four years (IQR 2–7.75).

One-fourth of the patients received an ACEI/ARB2/ARNI treatment. Beta-blockers were given to 49.8% of patients and were more frequently used in patients with HF (*p* < 0.001). Mineral receptor antagonists (MRA) and SGLT2 inhibitors were used in a few patients (2.3% and 1.8%, respectively). Loop diuretic use was frequent. 

NT-proBNP and US-troponin T levels were measured in 88 and 63 of 217 patients, respectively. NT-proBNP levels were higher in the HF group (median 4480 vs. 1344 pg/mL, *p* < 0.001), while US-Troponin T levels were similar.

In patients with a history of HF, 41.4% had an HF hospitalization the previous year, and 39.1% had required ambulatory intravenous loop diuretic treatment. The mean left ventricular ejection fraction was 56.2 ± 10.8%. Most of the patients (66 patients, 76.7%) had HFpEF, while ten patients (11.6%) had HFrEF and ten patients (11.6%) had HFmrEF. The mean ejection fraction was 61 ± 5%, 45 ± 2%, and 33 ± 5%, respectively. Among patients with HFrEF, 90% were treated with beta-blockers and 30% with ACEI/ARB2 or ARNI. None of the patients received MRA or iSGLT2.

### 3.2. Outcomes

Hospitalization for HF and the need for ambulatory intravenous diuretics were frequent, especially in patients with HF (Table 2). There were no differences in the hospitalization rate due to impaired renal function or in the need for RRT, which was needed in 30.4% of the population. However, in patients with previous HF, hemodialysis was the primary RRT used (79,2%), whereas in patients without HF, peritoneal dialysis or kidney transplantation were more frequently used (*p* = 0.008). 

The composite of all-cause death or need for RRT (Figure 1A) was more frequent in patients with previous HF. Patients with previous HF had almost twice the mortality compared to those patients with no HF history (40.2% vs. 22.3%; *p* = 0.005) (Figure 1B). A history of HF was independently associated with the composite endpoint (Table 3). 

#### Patients without HF History

Of the 130 patients with advanced CKD and no history of previous HF, only 19 (14.6%) developed HF during the 2-year follow-up period. Patients who developed HF were older (77.2 ± 7.4 vs. 71.2 ± 13.6; *p* = 0.007), and there were no differences in sex and cardiovascular risk factors except for a higher prevalence of diabetes mellitus (68.4% vs. 40.5%; *p* = 0.024). There were no differences in extracardiac and cardiac comorbidities. Medical treatment was similar, except for a higher use of insulin and anti-vitamin K anticoagulants in patients with HF onset (16.2 vs. 47.4%, *p* = 0.002, and 6.3 vs. 26.3%, *p* = 0.016, respectively). Of the patients who developed HF, 15 (78.9%) required hospitalization due to HF, and 11 (57.9%) required ambulatory intravenous diuretic treatment (Table 2). NT-proBNP levels were reported in 30 patients. It was significantly higher in patients who developed HF (3070 [1130–5888] vs. 658 [336–1760] pg/mL; *p* = 0.035). No differences in Troponin T levels (*n* = 20) were observed. The all-cause death rate was numerically higher (36.8 vs. 19.8%, *p* = 0.1). The composite of death or renal replacement therapy (53.1% of patients), hospitalization due to renal cause (26.9%), and the need for renal replacement therapy (32.3%) were similar between groups.

## 4. Discussion

In this cohort of patients with advanced renal disease (stages 4 and 5), 40% had a history of HF. The presence of previous HF identified a subgroup of high-risk patients, which was significantly and independently associated with the composite endpoint of all-cause hospitalization or the need for renal replacement therapy (HR 95% CI 1.62 (1.04–2.52), *p* = 0.034). Nineteen patients without HF at baseline developed HF during the 2-year follow-up. These patients were older and more frequently had diabetes; all-cause death was numerically higher (36.8 vs. 19.8%, *p* = 0.1).

The prevalence of HF in patients with end-stage kidney disease has been estimated to be between 35.8% [10] and 44% [13], similar to our study. It is worth noting that very few studies have analyzed this group of patients since studies have focused on patients on renal replacement therapies [14] or in less advanced CKD stages [3].

Patients with a history of HF were older and, more frequently, female. The prevalence of traditional cardiovascular risk factors was extremely high in our cohort, with 98.1% of patients having hypertension and 86.6% having dyslipidemia. More than half of the patients had diabetes mellitus, which was significantly higher in patients with HF (70.1%). Although the prevalence we observed is higher than in other series [15], a similar rate of cardiovascular risk factors has been described in patients with advanced chronic kidney disease, particularly those requiring renal replacement therapy or those with diabetic kidney disease [16,17]. Older age and high prevalence of cardiovascular risk factors, particularly diabetes mellitus, might explain the higher prevalence of cardiovascular diseases in patients with previous HF. Non-cardiac comorbidities did not differ between the groups. Although patients with previous HF had a higher prevalence of coronary artery disease and diabetes mellitus, 76.7% had HF with preserved ejection fraction. This high proportion of patients with HFpEF is consistent with previous studies, which have also reported a high prevalence of HF with preserved ejection fraction [18]. However, our findings contrast with other studies, in which HF with preserved ejection fraction accounted for only approximately 38% of HF patients. It is worth noting, though, that the ejection fraction was not measured in almost 20% of HF patients [10].

RAS inhibitor use was low in our series, with only 26.7% of patients receiving it. In contrast to our data, a Swedish registry with 24,283 patients with HFrEF, nearly 10% of whom had advanced CKD (stages 4 and 5), showed that 66% used RAS antagonists [9]. Although RAS inhibitors have a class I indication in the latest HF guidelines [11], the same document advises using ACE inhibitors, angiotensin II receptor blockers, sacubitril/valsartan, and mineralocorticoid receptor antagonists are contraindicated or should be used with caution or seek specialist advice. However, in a recent review, Beldhuis et al. conclude that in patients with HFrEF and CKD stage 4, there is evidence of the safety and efficacy of SGLT2 inhibitors, and with less evidence, ACE inhibitors, vericiguat, digoxin, and omecamtiv mecarbil. There is a lack of data on efficacy and safety for any HFrEF therapies in CKD stage 5 (eGFR < 15 mL/min/1.73 m^2^ or dialysis) [19]. Finally, a recent study in patients with advanced and progressive CKD showed that discontinuing RAS inhibitors was not associated with significant between-group differences in the long-term rate of decrease in eGFR [20]. Thus, a more aggressive approach to using RAS inhibitors should be adopted, especially in patients with HF and reduced ejection fraction, where they have a class I level of evidence A [11].

Overall, patients with HF were a high-risk group, with frequent HF hospitalization and the need for ambulatory intravenous diuretics. This group had nearly four times more risk of having a new HF hospitalization or ambulatory endovenous diuretic treatment than patients without baseline heart failure. Little information is available on patients with advanced CKD. In the ARIC study 3, which included more than 14,000 patients with CKD, the relative risk of developing HF was 1.94 in patients with eGFR < 60 mL/min/1.73 m^2^ compared to patients with FG > 90 mL/min/1.73 m^2^. However, only 2.7% (*n* = 403) had an eGFR < 60 mL/min/1.73 m^2^. In an analysis of three community-based cohort studies [21], the adjusted risk difference (95% CI) for HF was 4.6 (2.4, 6.7); *p* < 0.001. However, again, only 2.8% of patients had eGFR < 45 mL/min/1.73 m^2^. Kottgen et al. reported that the incidence of de novo HF was between 17 to 21%, similar to the 15% of patients without HF at baseline that developed HF in our study. The only differences between patients who developed HF and those who did not were older age and diabetes mellitus. Therefore, patients with diabetes and older patients should be identified as extremely high-risk. All-cause death was numerically higher (36.8 vs. 19.8%, *p* = 0.1). However, this data should be interpreted cautiously as only 19 patients developed HF during follow-up. 

The composite endpoint of all-cause death and the need for renal replacement therapies was 66.7% in the previous HF group compared with 53.1%, *p* = 0.046. The presence of previous HF was independently associated with the composite endpoint in multivariable analysis (HR 95CI 1.62 (1.04–2.52), *p*= 0.034). Studies on mortality in patients with advanced CKD have focused on patients undergoing dialysis or requiring a kidney transplant. The two-year survival rate reported in Medicare patients with HF and end-stage renal disease that received dialysis or had a functioning kidney transplant was 60.8% and 81.1%, respectively. This survival rate was lower than in patients without HF (76.9% and 92%, respectively) [10]. The mortality rate was also high in patients not undergoing renal replacement treatment. In the Swedish Register of patients with HF, the one-year mortality rate in patients with eFGR < 60 mL/min/1.73 m^2^ was about 23%, reaching 67% in HFpEF patients at the 5-year follow-up [9]. In our study, the two-year survival rate in patients with and without previous HF was 59.8% and 79.8%, which is higher than in patients without advanced CKD but lower than in patients already on renal replacement therapy.

Although the use of any renal replacement therapy was similar between groups, the type of therapy used differed. Notably, in patients with previous HF in whom renal replacement was indicated, only 20.8% were started on peritoneal dialysis. However, this method should be considered a first-line treatment in patients with HF [22]. Overall, this excess of risk highlights the need for specialized cardiorenal units, where cardiologists and nephrologists work together to improve the management and treatment of these patients [23,24].

Finally, previous studies showed that elevated NT pro-BNP and hs-TnT levels were significantly associated with cardiovascular events and higher HF risk [25,26]. Consistent with these studies, we found that NT-proBNP levels in patients with previous HF were significantly higher than those without HF. Moreover, patients who developed HF during follow-up also had significantly higher NT-proBNP levels. Nonetheless, only 40% and 30% of patients in our study had available NT-proBNP and US-troponin T levels, respectively.

### Limitations

The main limitation of our study is that it was a retrospective study, which means that we relied on data collected for clinical purposes rather than for research purposes. This meant that some important data, such as cardiac biomarkers and echocardiography, were unavailable for some patients, which may have limited our ability to characterize the study population fully. Additionally, as a hospital-based registry, there is a risk of selection bias, as patients who were not considered candidates for renal replacement therapies due to age, frailty, or comorbidities may not have been included in our study.

It is important to note that at the time of our study, the use of SLGT2 inhibitors for treating heart failure or chronic kidney disease was not yet approved. As a result, the number of patients in our study taking these medications was extremely low.

Despite these limitations, we believe that our study provides valuable insights into the prevalence and clinical characteristics of heart failure in patients with chronic kidney disease and adds to the growing body of literature on this crucial topic.

## 5. Conclusions

Patients with advanced CKD who are not on renal replacement therapy have a high prevalence (40%) and incidence (15%) of HF. Patients with a history of HF have a remarkably high prevalence of cardiovascular risk factors and cardiovascular comorbidities such as myocardial infarction or atrial fibrillation. The prognosis was poor, with a significantly higher composite of all-cause death or need for renal replacement therapy in patients with previous HF (66.7%, vs. 53.1 %, HR (95% CI) 1.62 (1.04–2.52), *p* = 0.034). For this reason, multidisciplinary management involving cardiologists and nephrologists is critical for improving these patients’ prognoses.

## Figures and Tables

**Figure 1 jcm-12-02339-f001:**
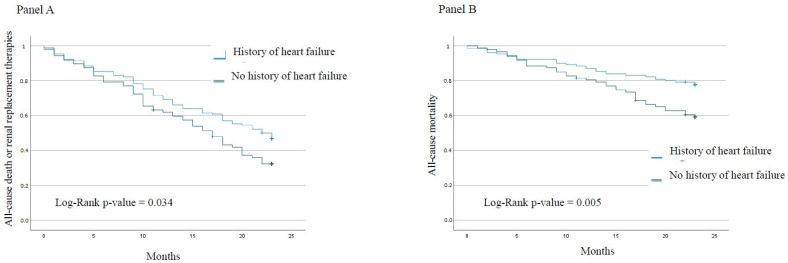
Kaplan–Meier curves for long-term outcome divided by a history of HF. (**Panel A**): All-cause death or renal replacement therapies. (**Panel B**): All-cause mortality.

**Table 1 jcm-12-02339-t001:** Baseline characteristics between patients with and without heart failure at inclusion.

	No Heart Failure *n* = 130 (60%)	Heart Failure *n* = 87 (40%)	*p*-Value
Baseline Characteristics			
Age (years)	72.1 ± 13.1	78.2 ± 8.8	<0.001
Women	47 (36.2)	46 (52.9)	0.015
Hypertension	128 (98.5)	85 (97.7)	0.683
Diabetes mellitus	58 (44.6)	61 (70.1)	<0.001
Dyslipidemia	109 (83.8)	79 (90.8)	0.14
BMI	28.0 ± 5.1	29.6 ± 5.7	0.016
Never smoker	84 (64.6)	57 (65.5)	0.25
Active smoker	25 (19.2)	22 (25.3)	
Previous smoker	21 (16.2)	8 (9.2)	
Stroke/TIA	16 (12.3)	13 (14.9)	0.58
Sleep apnea	15 (11.5)	17 (19.5)	0.10
COPD/Asthma	19 (14.6)	19 (21.8)	0.17
Peripheral vascular disease	31 (23.8)	24 (27.6)	0.54
Cancer	37 (28.5)	19 (21.8)	0.28
Myocardial infarction	13 (10.0)	26 (29.9)	<0.001
Percutaneous coronary intervention	7 (5.5)	23 (26.4)	<0.001
Moderate-severe valve disease	3 (2.3)	15 (17.2)	<0.001
Atrial fibrillation/flutter	14 (10.8)	48 (55.2)	<0.001
Baseline Treatment			
ACEi/ARB2/ARNI	40 (30.8)	18 (20.7)	0.1
Beta blockers	52 (40.0)	56 (64.4)	<0.001
MRA	4 (3.1)	2 (2.3)	0.74
iSLGT2	1 (0.8)	3 (3.4)	0.15
Insulin	27 (20.8)	40 (46.0)	<0.001
Other oral anti-diabetic drugs	24 (18.5)	23 (26.4)	0.16
Antiplatelet therapy	43 (33.1)	36 (41.1)	0.21
Statins	98 (75.4)	68 (78.2)	0.64
Loop diuretic	66 (50.8)	78 (89.6)	<0.001
HCTZ/higrotone	7 (5.4)	14 (16.1)	0.009
Anti-vitamin K	12 (9.2)	21 (24.1)	0.003
Direct-acting anticoagulants	1 (0.8)	14 (16.1)	<0.001
Intravenous iron	21 (16.2)	20 (23.0)	0.21
Oral iron	75 (57.7)	56 (64.4)	0.32
Erythropoietin	52 (40.0)	46 (52.9)	0.062
Laboratory results			
Creatinine (mg/dl)	3.5 ± 2.2	3.0 ± 1.0	0.005
eGFR (mL/min/1.73 m^2^)	17.7 ± 5.5	18.4 ± 5.5	0.18
Potassium (mmol/L)	4.8 ± 0.5	4.6 ± 0.6	<0.001
Creatinine/protein ratio (mg/g)	1008.1 (362.0 − 2184.0)	663.0 (286.1–1859.0)	0.45
HbA1c (%)	6.4 ± 1.3	6.6 ± 1.4	0.15
Hemoglobin (g/dl)	12.1 ± 1.6	12.3 ± 1.9	0.16
Transferrin saturation index (%)	26.4 ± 10.4	24.7 ± 8.7	0.11
LDL cholesterol (mg/dl)	87.3 ± 27.1	78.7 ± 31.7	0.02
NT-proBNP (*n* = 88) (ng/dl)	1344 (393−3721)	4480 (1417−7755)	<0.001
HS-Troponin T (*n* = 63) (ng/L)	47.0 (34.3−57.3)	52.0 (31.7−73.8)	0.61

Data are presented as numbers and percentages, mean and standard deviation, or median and interquartile ranges, as appropriate. Abbreviations: BMI: body mass index. TIA: transient ischemic attack. COPD: chronic obstructive pulmonary disease. ACEi: Angiotensin-converting-enzyme inhibitors. ARB2: Angiotensin II receptor blockers. ARNI: angiotensin receptor-neprilysin inhibitor. MRA: mineralocorticoid receptor antagonists. iSLGT2: sodium-glucose cotransporter type 2 inhibitors. HCTZ: hydrochlorothiazide. EGFR: estimated glomerular filtration rate. LDL: low-density lipoprotein. NT-proBNP: N-terminal-pro hormone blood natriuretic peptide. HS troponin T: high-sensitivity troponin T.

**Table 2 jcm-12-02339-t002:** Outcomes of patients with heart failure at inclusion compared to those without HF.

	No Heart Failure (*n* = 130, 60%)	Heart Failure (*n* = 87, 40%)	*p*-Value
Heart failure hospitalization, *n* (%)	15 (11.5)	35 (40.2)	<0.001
Ambulatory intravenous diuretic, *n* (%)	11 (8.5)	39 (44.8)	<0.001
Hospitalization due to renal cause, *n* (%)	35 (26.9)	20 (23.0)	0.51
Renal replacement therapies, *n* (%)	42 (32.2)	24 (27.6)	0.46
Hemodialysis, *n* (%)	18 (42.9)	19 (79.2)	0.008
Peritoneal dialysis, *n* (%)	16 (38.1)	5 (20.8)	
Kidney transplant, *n* (%)	8 (19.0)	0 (0)	
All-cause death, *n* (%)	29 (22.3)	35 (40.2)	0.005
All-cause death or renal replacement therapy, *n* (%)	69 (53.1)	58 (66.7)	0.046

**Table 3 jcm-12-02339-t003:** Multivariate Cox proportional model for the hazard ratio of 2-year all-cause mortality or need for renal replacement therapies.

	Adjusted HF (97%CI for HR)	*p*-Value
Female	0.82 (0.56–1.21)	0.32
Age, per year	1.003 (0.99–1.022)	0.74
Previous HF	1.62 (1.04–2.52)	0.034
Diabetes mellitus	0.72 (0.49–1.05)	0.09
Myocardial infarction	1.72 (1.08–2.73)	0.022
Moderate-to-severe valve disease	0.98 (0.51–1.88)	0.95
Atrial fibrillation	1.03 (0.66–1.62)	0.90
eGFR, per mL/min/1.73 m^2^	0.87 (0.84–0.90)	<0.001
RAAS inhibitors	1.19 (0.76–1.87)	0.44
Beta-blockers	0.89 (0.61–1.31)	0.55
Loop diuretics	1.21 (0.78–1.86)	0.40

Abbreviations: HF: heart failure. eGFR: estimated glomerular filtration rate. RAAS inhibitors: Renin angiotensin aldosterone system inhibitors.

## Data Availability

The data underlying this article will be shared on reasonable request to the corresponding author.

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
