# Peer review of "Characteristics of Patients with Heart Failure and Advanced Chronic Kidney Disease (Stages 4–5) Not Undergoing Renal Replacement Therapy (ERCA-IC Study)"

_jcm, 2023, doi:10.3390/jcm12062339_

Round 1

Reviewer 1 Report

This is an interesting study attempting to estimate the prevalence and consequences of heart failure among patients with stage 4/5 chronic kidney failure, demonstrating that HF diagnosis is a marker of increased risk of mortality and the need of RRT among such patients. The concept is interesting, the cohort is moderately large. I have some comments/questions:

1. Since heart failure is the major point of interest here, it must be defined, i.e. what were the criteria of heart failure diagnosis? How was HF classified to HFpEF, HFrEF and HFmrEF? The authors should emphasize that more than 3/4 of their HF patients had HFpEF.

2. Would it be possible to include time from HF diagnosis in the model (i.e. how long HF persisted before the inclusion of a patient in the analysis)?

3. English needs correction (i.e. Eritropoyetin )

Author Response

Response to Reviewer 1 Comments

Point 1: Since heart failure is the major point of interest here, it must be defined, i.e. what were the criteria of heart failure diagnosis? How was HF classified to HFpEF, HFrEF and HFmrEF? The authors should emphasize that more than 3/4 of their HF patients had HFpEF.

Response 1:

We want to thank the reviewer for this important comment. The diagnosis of heart failure was made per the European Society of Cardiology’s HF guidelines (2021). It is widely recognized that distinguishing between true heart failure and volume overload can be a clinical challenge in patients with advanced chronic kidney disease. However, there are currently no specific diagnostic criteria for this patient population.

HF with preserved ejection fraction (EF) (HfpEF) was defined as an EF greater than or equal to 50%, HF with reduced EF (HfrEF) as an EF less than or equal to 40%, and HF with mildly reduced EF (HFmrEF) as an EF between 41% and 49%.

We have added this information in the Methods section and emphasized in the discussion the high proportion of HFpEF found in our study.

Point 2:  Would it be possible to include time from HF diagnosis in the model (i.e. how long HF persisted before the inclusion of a patient in the analysis)?

Response 2: The median time from diagnosis of heart failure to inclusion in the study was 4 years (IQR 2-7,75).

Point 3: English needs correction (i.e. Eritropoyetin )

Response 3:  We have reviewed the manuscript and have completed a thorough proofreading of the text to ensure that it is free from any spelling and grammatical errors. We believe these revisions have improved the clarity and readability of the manuscript.

Reviewer 2 Report

1) in the introduction or the discussion there should be more discussion of the different types of cardiorenal syndrome. Type one through four.

2) the strength of the limitations of the study should be reported

3) it is likely that many of the patients with advanced heart failure chose conservative management and that’s why the renal replacement therapy incidence  was not different between the two groups that should be discussed and if you have data supporting this it should be presented

Author Response

Response to Reviewer 2 Comments

Point 1: In the introduction or the discussion there should be more discussion of the different types of cardiorenal syndrome. Type one through four.

Response 1:  Thank you for this comment. We have included in the introduction the classification of the different types of cardiorenal syndrome. We have also clarified how heart failure was defined.

Point 2:  The strength of the limitations of the study should be reported.

Response 2: We have expanded the limitations section of the manuscript, with a particular emphasis on highlighting the strength of the limitations of our study.

Point 3: It is likely that many of the patients with advanced heart failure chose conservative management and that’s why the renal replacement therapy incidence  was not different between the two groups that should be discussed and if you have data supporting this it should be presented

Response 3:  Thank you for this comment. The incidence of renal replacement therapy did not differ between the two groups. Although patients with previous HF were older and had a higher frequency of diabetes, both groups had a high prevalence of cardiovascular risk factors and cardiovascular disease, which may have influenced the progressive decline in eGFR and the finally need of RRT. We did not collect data on the reasons why renal replacement therapy was not pursued in some patients. However, it is worth noting that the type of RRT was different in both groups, likely reflecting the different preferences or circumstances of the patients. We have highlighted that in the text.

Round 2

Reviewer 2 Report

The reviewers comment where addressed